# Assessment of a Novel Stress and Immune Gene Panel on the Development of Australasian Snapper (*Chrysophrys auratus*) Larvae

**DOI:** 10.3390/genes15121520

**Published:** 2024-11-27

**Authors:** Kerry L. Bentley-Hewitt, Duncan I. Hedderley

**Affiliations:** The New Zealand Institute for Plant and Food Research Limited, Private Bag 11600, Palmerston North 4442, New Zealand; duncan.hedderely@plantandfood.co.nz

**Keywords:** aquaculture species, snapper, gene expression, larvae, stress immune

## Abstract

Background: Larvae development is a critical step in aquaculture, yet the development of immune and stress responses during this early phase of life is not well understood. Snapper is a species that has been selected as a candidate for aquaculture in New Zealand. Methods: In this study we explore a set of 18 genes identified as potentially being involved in the stress and immune responses of snapper larvae during the first 30 days of development. Larvae were collected between 11:45 a.m. and 16:10 p.m. each day. Results: Most genes did not deviate from baseline expression throughout the 30 days, with some exceptions between Days 0 and 6 with glyceraldehyde-3-phosphate dehydrogenase and superoxide dismutase, mitochondrial uncoupling protein 2-like, peroxiredoxin-5 mitochondrial, and hepcidin, which predominantly increased and then stabilized by Day 6 until Day 30. Some genes were affected by the time of day, such as actin cytoplasmic 1 and catalase isoform X2. Conclusions: This exploratory study is the first to look at a panel of stress- and immune-related marker gene expression during early snapper development. It sets methods in place to explore the expression of these markers and determine the impact of different potential stressors, such as alternative food sources and other environmental changes. It also highlights the importance of same time of day collections for gene expression studies.

## 1. Introduction

Larvae development is a critical step in aquaculture, yet the development of immune and stress responses during this early phase of life is not well understood, even though these processes are critical to survival.

Stress responses involve the neural, endocrine, and immune systems and can be acute (increasing innate responses) or chronic (immune-suppressive and increasing the risk of infection) [1]. The majority of teleost species are ectothermic, and immune responses can depend on seasonal changes, such as temperature increases causing enhanced immune responses and temperature decreases suppressing responsiveness to antigens [2]. Research into the immune systems of fish have been limited by reagents, such as antibodies, for classical immunology studies [3]. However, with the availability of a species transcriptome, gene expression can be used to investigate potential biomarkers that are indicative of a fish’s health status. Biomarkers are crucial tools for assessment of environmental changes [4]. Fish show conservation of most immune-related genes with vertebrates, but investigations into functional immunology remain scarce [5].

Antioxidant enzymes are often used in toxicology as sensitive biomarkers, e.g., superoxide dismutase (SOD) and catalase (CAT), which constitute the first line of defence against reactive oxygen species. Aside from the antioxidant enzyme genes themselves, another group of genes involved in the immune response is induced under oxidative stress conditions. These can influence susceptibility to diseases and includes antimicrobial peptide, such as hepcidin (HAMP). The great sensitivity to a wide range of stress conditions makes the gene expression of these antioxidant enzymes and immune-related genes suitable for use as fish biomarkers [4].

The Australasian snapper, *Chrysophrys auratus*, tāmure, (family Sparidae, hereafter referred to as snapper) is a species that has been selected as a candidate for aquaculture in New Zealand and a selective breeding program was started in 2016 [6,7]. Snapper is closely related to *Pagrus/Chrysophrys major* (hereafter referred to as red seabream), a major aquaculture species in Japan [8], and is a valuable commercial and recreational fish species located around the coasts of Australia and New Zealand [9]. Breeding of snapper in New Zealand started around 20 years ago at The New Zealand Institute for Plant and Food Research Limited (Plant & Food Research) in Nelson, and in 2016 the programme began using genomic information to investigate and select for traits of economic interest [10]. Research at Plant & Food Research has included the development of diverse tools, such as a genome, linkage map, transcriptome, and genome-wide sequence information of pedigreed snapper, which have been used to improve breeding of snapper [10,11,12].

A panel of genes was previously used to determine the impacts of temperature changes in snapper fin, liver, and head kidney tissues. The details of the gene targets, how they were selected and their potential link to fish stress and immune responses, were previously published in Bentley-Hewitt et al. [13].

Exploration into immune- and stress-related genes in larvae has been previously studied in a related species to snapper, *Sparus aurata* (hereafter referred to as gilthead seabream). Expression of some genes related to stress and immune response in early larvae phase (30 days post-hatch) were modified in offspring of broodstock that were fed diets with differing amounts of linseed oil, which replaced the standard fish oil in the diet, thus demonstrating that early gene modification in larvae could be used to monitor the tolerance of larvae to new diet changes [14].

In this study we explore a set of genes identified as potentially being involved in the stress and immune responses of snapper larvae during the first 30 days of development.

## 2. Materials and Methods

### 2.1. Egg Collection, Larvae Growth, and Larvae Rearing

Larvae collections were conducted at Plant & Food Research’s Finfish facility in Nelson. The samples for this experiment were taken from a single clear 200 L polycarbonate tank. This tank was located in an insulated room in the Finfish facility hatchery, which was maintained at temperature (details below) by a Toshiba 7Kw heat pump inverter. The room also had ceiling-mounted fans for further air circulation. The influent to the tank was gravity fed from a 7000 L plastic silo (Advantage Plastics, New Zealand). Silo water was temperature-controlled using four HX-8 heat exchangers in a parallel arrangement (Vaportec Spirex, New Zealand) and aerated with four AQS385 air diffusers. This silo was elevated in a steel frame and located outside the temperature-controlled room. The influent water was also UV treated and filtered to 1 µm, by a process of first transferring through a set of 5 µm bag filters contained in the X100 casings (Sefar, Heiden, Switzerland), then through a set of 1 µm cartridge filters contained in Big Bubba housings (Watts filters, North Andover, MA, USA). Sea water then flowed through the Emperor Safeguard 300 Watt UV system (Pentair, Golden Valley, MN, USA) for sanitisation. Effluent water from the experiment tank passed through a small purpose-built nylon mesh tambourine style filter before going to waste. The effluent filter started with 125 µm mesh and was changed to a 250 µm mesh as the larvae grew. Gentle aeration was provided by bubbling compressed air through a small cedar airstone at the bottom of the tank. Lighting in the temperature-controlled room was provided by Aqua Shift, 14 Watt LED light fixtures. These lights were programmed to mimic the natural photoperiod at the time of the experiment (15:9 light–dark hours).

Twenty grams of washed snapper eggs were added to the tank with very gentle aeration provided and an influent flow of 0.25 lpm. The influent water and room temperature were set to replicate the water temperature of the tank from which the eggs were collected. This water was initially 18 °C, then the temperature was raised over the subsequent 9 days to 22 °C, then maintained at 22 °C +/− 1 °C over the course of the experiment. Dissolved oxygen readings were taken twice a day and were maintained in the range of 6–9 mg/L over the course of the experiment.

The eggs hatched the day after being put in the tank and this was recorded as Day 0 for the experiment. Snapper larvae hatch with a yolk sack, which provides nutrition for their initial 3–4 days until the larvae’s mouth opens. On Day 3 (3 days post-hatch) a light background feed of *Nannochloropsis* algae for rotifers was added to the tank along with a first feed of rotifers (*Brachionus plicatilis*). These rotifers were enriched with S.presso (INVE Technologies, Dendermonde, Belgium) and the target density of rotifers in the tank was 10–15 rotifers/mL. On Day 23 the larvae were sufficiently developed to start receiving artemia (*Artemia franciscana*). The artemia were fed in batch editions, to allow equal opportunity for all the larvae to access them. Around Day 26 the first inert feed of O.range Start-S (INVE Technologies, Dendermonde, Belgium) was sparingly introduced to the fish. The larvae tank was cleaned by siphoning the sides and bottom with a ‘broom siphon’ every few days. The cleaning performed was altered depending on the larvae stage and feed type.

Triplicate samples of 25 mg of larvae were collected every second day for analysis. A novel funnel and mesh system was required to collect these larvae because the collection of “dry” larvae of this age and size was a unique challenge (see Figure 1). At each step, every endeavour was made to reduce larval stress.

The eggs/larvae were netted from the tank then lightly sedated using Aqui-S (Aqui-S New Zealand Ltd., New Zealand) anaesthetic at a concentration of at 40 mg/L in water [15]. Once sedated the larvae were gently poured through this funnel system and captured on a pre-weighed patch of mesh. This mesh was then dabbed dry and reweighed to achieve a minimum net weight of 25 mg of larvae. The larvae were then sluiced off this mesh using RNAlater (Invitrogen, Carlsbad, CA, USA) directly into a 1.5 mL micro tube. The tube was then topped up with RNAlater and refrigerated at 4 °C overnight. This procedure was performed in triplicate. The following day the samples were transferred to a −80 °C freezer. This sample collection procedure was repeated every second day from Day 0 to Day 30 producing 16 triplicate sample sets. Images of Day 2 and Day 30 snapper larvae are shown in Appendix A.

### 2.2. Gene Target Sequence Design

Candidate genes were selected from a literature review of studies looking at immune- and stress-related genes, mainly in red and gilthead seabream, and whether they could potentially be identified in the snapper transcriptome. Details of the literature searches and the design process were applied according to Bentley-Hewitt et al. [13].

### 2.3. RNA Extraction

Larvae (approx. 25 mg) were thawed and blotted to remove excess RNAlater. Larvae were then homogenized at maximum speed for 20 s with an OMNI THQ TM homogenizer (OMNI International, Kennesaw, GA, USA) in 600 µL RLT buffer (Qiagen, Hilden, Germany) with 6 µL beta-mercaptoethanol (Sigma-Aldrich, Auckland, New Zealand). The samples were centrifuged at 17,000 g for 5 min at room temperature and the supernatant was transferred to a new tube. An equal volume of 70% molecular grade ethanol was added to the supernatant and transferred to a Qiagen RNeasy MINI column (Qiagen, Hilden, Germany). RNA was extracted as per the manufacturer’s protocol (RNeasy mini handbook, Fourth edition, June 2012) into 50 µL of water for gene expression analysis of 48 genes (Table 1) by the Counter Analysis System (NanoString Technologies, Seattle, WA, USA). RNA quantity was assessed using the Qubit™ RNA BR (Broad-Range) Assay Kit as per the manufacturer’s instructions (Invitrogen™, cat. no. Q10211, Carlsbad, CA, USA) and a Qubit^®^ 2.0 Fluorometer (Invitrogen™, Carlsbad, CA, USA).

### 2.4. Gene Expression Analysis—NanoString nCounter Analysis System

Larvae RNA samples (28 ng) were analysed using the nCounter Plexset reagents (NanoString, Seattle, WA, USA) following the manufacturer’s instructions. Target sequences were designed by NanoString Technologies, Inc. and ordered from Integrated DNA Technologies, Inc., Coralville, IA, USA. Details of target sequences are shown in Table 1. Raw gene counts were analysed using the NanoString nCounter Analysis System as described previously [13]. Raw counts were normalized using the positive controls, and target genes were normalized to the internal reference genes: 40S ribosomal protein S18 (*rps18*), 60S ribosomal protein L8 (*rpl8*), hypoxanthine-guanine phosphoribosyltransferase (*hprt1*), and elongation factor 1-alpha (*ef1a*), using nSolver™ 4.0 analysis software (NanoString, Seattle, WA, USA).

### 2.5. NanoString Efficiency

This novel panel of snapper genes was previously designed for another project to explore the effects of temperature on juvenile snapper [13]. The expression of genes varied considerably and to avoid overloading of the NanoString cartridges the concentration of RNA added was low, with our previous study using 800 and 1450 ng of organ tissue [16]. This resulted in many genes on the panel having low gene counts which could not be calibrated by the assay’s nSolver™ software, as each gene requires 50 counts in the eight lanes containing the calibration sample. These genes had to be removed from the analysis, leaving 18 target genes (44 target genes were tested in total) and 4 reference genes. Glyceraldehyde-3-phosphate de-hydrogenase (*gapdh*) was removed as a reference gene because of its variability across samples, and was treated as a target gene instead. The final target genes were normalized to four reference genes that were stable and passed calibration.

### 2.6. Statistical Analysis

For each gene, the average expression count for each day was calculated. To explore how gene expression profiles varied over time, a principal components analysis was carried out, using the correlations between gene counts. Collection day was included as a supplementary categorical variable. To explore whether time of day may have affected the expression level of genes, we fitted a cubic polynomial trend over days to the expression data and looked to see if the time of day that larvae were sampled was associated with variations from that trend. The analyses were carried out using R (version 4.0.5, The R Foundation for Statistical Computing) using packages stats (version 4.0.5, The R Foundation for Statistical Computing), ggplot2 (version 3.3.6, Wickham [17]), and FactoMineR (version 2.6, Husson et al. [18]).

## 3. Results

### Larvae Gene Expression

For the 18 target genes detected, most genes had stable levels of expression throughout the 30 days: *gapdh* and *sod1* rose by about a factor of 10 between Days 0 and 2; *ucp-2* like and *prdx5* were not present at Day 0, but were from Day 2 onwards, and *hamp* varied in presence between Days 0 and 6, then stabilized after Day 6 (Figure 2).

One way to summarize the patterns in expression data was to use a principal components analysis (PCA). The PCA on correlations between the genes for each day showed that Day 0 (orange ellipse) clustered differently than other days. Days 2–6 (yellow ellipses) also clustered differently. By Days 8–10 (green ellipses) the expression of genes was much closer to the main cluster, which all clustered together (Figure 3). A table showing the variables that correlate with the first five principal components (PCs) are included in Appendix A.

A potential limitation of this study was the slight variance in the time of day that samples were collected, with the earliest collection time being 11:45am and the latest being 16:10pm. To explore whether time of day may have affected the expression level of genes, we fitted a trend over days to the expression data and looked to see if the time of day that larvae were sampled was associated with variations from that trend (Figure 4). For *actb*, *cat,* and *prdx1*-like, there do seem to be effects, with samples taken earlier in the day having lower values (*p*-values for adding a linear effect for time of day to the polynomial trend over collection day between <0.001 and 0.018; R-squared for time of day effect between 5% and 28%); for *gsr* and *nrf2*, there seems to be the opposite effect, with samples taken earlier in the day having higher expression (*p*-values between 0.001 and 0.023; R-squared for time of day effect between 10% and 13%). Figure 4 shows the raw expression data with trend lines and points colour coded to the time of day collected (darker colours being earlier in the day). This shows that *cat* expression does appear to have clear differences between earlier and later collections, with earlier time points having lower expression levels. In Figure 4, the associations with time of day and expression levels of *actb*, *prdx1-like*, *gsr,* and *nrf2* do not appear as clear cut as with *cat* expression (time of day accounted for 28% variance in *cat* and 5–13% for *actb*, *prdx1-like*, *gsr,* and *nrf2*).

## 4. Discussion

The results of this study showed that most genes tested in larvae between Days 0 and 30 remained relatively stable, although there were a few exceptions with very early day samples with *gapdh*, *sod1*, *ucp2-like*, *prdx5,* and *hamp*. Whole-body larvae gene expression could be a limiting factor because tissue-specific effects may get buried by overall gene expression. However, a study in gilthead seabream whole larvae (30 days post-hatch) found significant daily variations in expression of genes from poorly representative tissues such as the pineal gland, along with genes expressed ubiquitously [19]. Juvenile snapper fin, head, kidney, and liver were analysed with the panel of genes presented in this study, however, six more genes passed calibration in these tissues (*muc18-like*, *c8a*, *igf1*, *ptgs2*, *soc3*, and *tgfb1-like*), indicating that expression of those genes was very low or not present in snapper larvae in the first 30 days of development [13]. For example, *muc18-like,* or MUC18 (the expected protein form), is a predominant mucin in skin, gills, and stomach in gilthead seabream and offers a first line of defence [20]. Therefore, it may not be being produced at high concentrations whilst snapper are in the early development stage. Another first line of defence is the complement system. C8A is a terminal complement protein, which forms the membrane attack complex and can lyse certain pathogens and cells [21]. The *c8a* and *muc18-like* genes are both not present or not highly expressed in larvae, which demonstrates that these immune defences may not be well established in the first 30 days of larvae development. However, another component of the complement system, *C3-like*, was found in larvae in this study, indicating that components of the system are at least partly present. A study in Eurasian perch larvae demonstrated a difference in *c3* gene expression when comparing domesticated (higher expression) to wild Eurasian perch in early developmental stages (18–20 days post-hatch). In addition, domesticated fish coped better with thermal stress [22]. This indicates the potential importance of this gene in the tolerance of larvae to thermal stress.

This study also highlighted the importance of collecting samples at the same time of day, since the gene expression of many genes, including antioxidant response genes, fluctuated in daily rhythms along with the circadian genes, such as *cry1,* used in our study. For our study, the collection time of samples did vary each day between 11:45 a.m. and 16:10 p.m. We explored whether this small variation in collection time may have affected our levels of expression by plotting earlier and later collection times against the trendline of expression over the days. Most convincing was the effect of time of collection with *cat* expression where earlier times had lower expression (Figure 4). Changes in expression of some genes related to stress and immune response in the early larvae phase (30 days post-hatch) were modified in offspring of broodstock fed diets with differing amounts of linseed oil with replaced fish oil [14]. For example, *hsp 70* (a key gene for cytoprotection) increased approximately 1.5-fold in larvae from parents fed linseed oil, which replaced fish oil by 60, 80 and 100%. Whilst pro-inflammatory Tumour Necrosis Factor alpha (*tnf-alpha*) increased over 2-fold in larvae from parents fed 80% linseed oil compared with those fed 0, 60 and 100% linseed oil, which replaced fish oil in their diets.

Future studies should consider analysis of gene biomarkers using at least two different time points to help understand the circadian rhythmicity of many genes, but as a minimum the time of day should be very consistent. For example, gilthead seabream larvae have low levels of reactive oxygen species via increases in nuclear factor erythroid 2–related factor 2 (*nrf2*) mediated stress response pathways including glutathione S-transferase (*gst*) and heme oxygenase during the feeding period and early night, whilst *hsp 70* and *hsp90* were at their maximum expression during the second half of feeding [19]. This highlights that, dependant on the time of day, genes may be expressed differently, so consistent collection times are critical for gene expression studies.

Another limitation of this exploratory study was that samples were collected from just one tank, therefore, there could be an unknown tank factor influencing expression, such as a bacterial contaminant that can be introduced with feeds. However, this exploratory study is the first to look at a panel of stress- and immune-related marker gene expressions during early snapper development and sets methods in place to explore the expression of these markers and determine the impact of different potential stressors, such as alternative food sources and other environmental changes. Future work should focus on demonstrating which of the gene markers are useful to predict stress or disease responses in the larvae and determine the stressors that cause the most significant effects on gene changes. This approach could potentially be used as a fish health husbandry management tool.

In conclusion, this study is the first to explore immune- and stress-related genes in snapper larvae. It indicates the components of the immune system and stress response related genes are present in very early development of snapper larvae. Understanding these processes is critical to understanding fish health and can aid the development of good aquaculture practices for the species.

## Figures and Tables

**Figure 1 genes-15-01520-f001:**
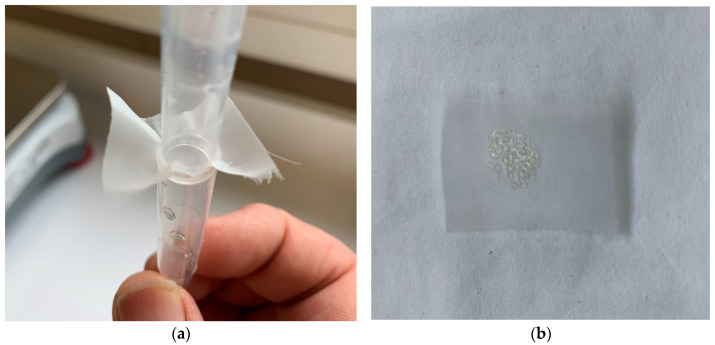
The funnel and mesh arrangement used to capture the “dry” eggs/larvae of *Chrysophrys auratus*, tāmure, (family Sparidae, referred to as snapper) (**a**). The pre-weighed patch of mesh with collected snapper eggs (**b**).

**Figure 2 genes-15-01520-f002:**
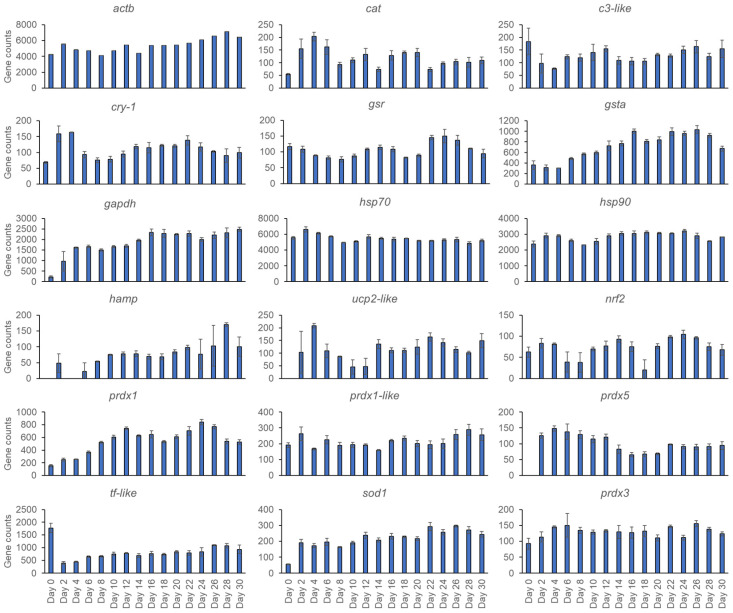
Mean counts (*n* = 3) for *Chrysophrys auratus*, tāmure, larvae genes (family Sparidae, referred to as snapper) +/− standard errors collected every 2 days until 30 days and including Day 0, which was a hatched egg.

**Figure 3 genes-15-01520-f003:**
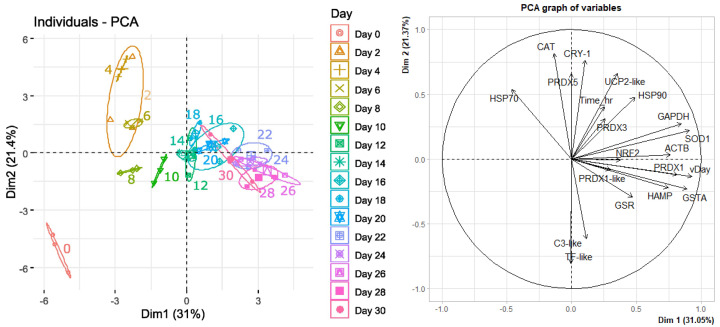
Mean counts (*n* = 3) for *Chrysophrys auratus,* tāmure, larvae genes (family Sparidae, referred to as snapper) every 2 days until 30 days are shown as individual data points and colour coded for each day on the principal components analysis (PCA) graph of individuals, whilst variables influencing the individual data points are shown on the PCA graph of variables.

**Figure 4 genes-15-01520-f004:**
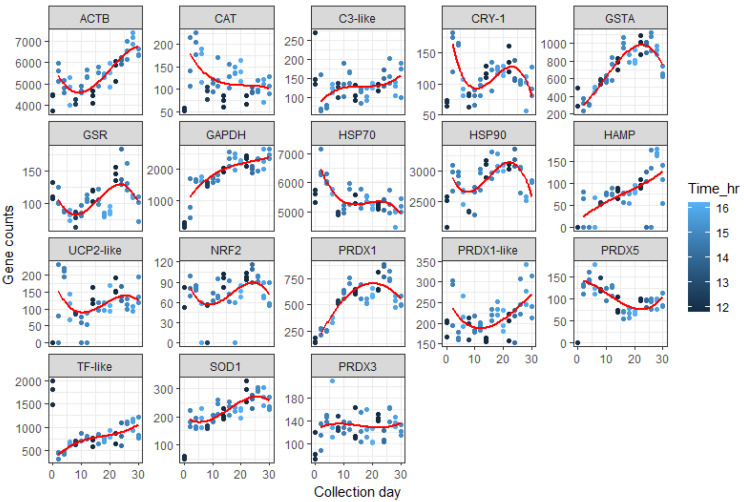
Mean counts (*n* = 3) *Chrysophrys auratus*, tāmure, larvae genes (family Sparidae, referred to as snapper) collected every 2 days until 30 days, fitted with trend lines and points colour coded to the time of day collected (darker colours being earlier in the day).

**Table 1 genes-15-01520-t001:** Gene targets analysed by NanoString Technologies Inc. (Seattle, WA, USA).

Gene Name	GenBank Accession Number	Target Region	Target Sequence	HUGO Gene
Reference genes				
Elongation factor 1-alpha (*ef1a*)	Ch_aur001.1	1152–1251	CAAGAAGCTTGAGGATGCTCCCAAGTTCGTCAAGTCTGGTGATGCCGCCATTGTCAAACTGCACCCACAGAAGCCCATGGTTGTGGAGCCCTTCTCCAGC	LOC115578802
60S ribosomal protein L8 (*rpl8*)	Ch_aur022.1	634–733	GGTGGTGGTAACCATCAGCATATTGGCAAACCCTCAACAATCAGAAGGGACGCACCTGCTGGTCGCAAGGTCGGTCTCATTGCTGCCCGTCGTACAGGCA	rpl8
40S ribosomal protein S18 (*rps18*)	Ch_aur036.1	184–283	GAGGTTGAGCGTGTGGTGACCATCATGCAGAATCCTCGCCAGTACAAAATCCCAGACTGGTTCCTCAACAGGCAGAAGGACGTCAAGGACGGCAAATACA	LOC115577508
Hypoxanthine-guanine phosphoribosyltransferase (*hprt1*)	Ch_aur031.1	253–352	CTGAACAGGAACAGTGACCGCTCCATCCCAATGACAGTGGACTTCATCCGCCTCAAGAGCTACTGTAACGACCAGTCGACAGGTGAAATCAAAGTGATTG	hprt1
Target genes				
Actin cytoplasmic 1 (*actb*)	Ch_aur069.1	736–835	CAGGTCATCACCATCGGCAATGAGAGGTTCCGTTGCCCAGAGGCCCTCTTCCAGCCTTCCTTCCTCGGTATGGAGTCCTGCGGAATCCACGAGACCACCT	actb
Catalase isoform X2 (*cat*)	Ch_aur025.1	641–740	GCTACGGCTCTCACACCTTCAAACTGGTCAATGCCAATGGTGAGCGTTTCTACTGCAAGTTCCACTACAAGACTGATCAAGGAATAAAGAATCTGACAGT	cat
Complement C3-like A (*c3-like*)	Ch_aur061.1	2871–2970	TCTGATTCTCAATGCACAGCAACCTGACGGCATGTTTAAAGAAGTTGGAACGGTCTCCCACGGGGAGATGATTGGCGATGTGCGCGGCGCAGATTCAGAT	LOC115582848
Cryptochrome-1-like (*cry1-like*)	Ch_aur034.1	1655–1754	ACCAACAAACCAGCATCGGAACACACCAGCAAGGTTATCCAGGTACCAGTGCCGGTGTGATGTGTTACACTCAAGGCACACCACAGCAGATTCCTGGTTC	LOC115595869
Glutathione reductase mitochondrial isoform X2 (*gsr*)	Ch_aur019.1	236–335	TCAATGTTGGCTGTGTCCCTAAGAAGGTTATGTGGAATGCTGCAAGTCACGCCGAGTATCTCCATGATCACAATGATTATGGCTTCGACGTTGGAAATGT	gsr
Glutathione S-transferase A (*gsta*)	Ch_aur021.1	284–383	AACTGGCAATGATGTACCAGCGCATGTTTGAGGGTCTCTCACTCAACCAGAAAATGGCGGATGTCATCTACTACAACTGGAAGGTCCCAGAGGGAGAGAG	LOC115579480
Glyceraldehyde-3-phosphate dehydrogenase (*gapdh*)	Ch_aur068.1	303–402	CTTGAAGGGTGGTGCCAAGAGAGTCATCATCTCTGCACCCAGCGCCGACGCTCCCATGTTTGTCATGGGTGTCAACCATGAGAAGTACGACCATTCCCTC	gapdh
Heat shock cognate 70 kDa protein (*hsp70*)	Ch_aur056.1	1605–1704	GGTGTCTGCTAAGAATGGCCTGGAGTCGTATGCTTTCAACATGAAGTCTACTGTGGAGGATGAAAAACTTGCTGGCAAAATCAGTGATGACGACAAGCAG	LOC115594641
heat shock protein HSP 90-beta isoform X2 (*hsp90ab1*)	Ch_aur007.1	514–613	GGAGCTGACATCTCCATGATTGGTCAGTTTGGTGTGGGTTTCTACTCTGCCTACCTTGTTGCTGAGAAGGTGGTCGTCATCACCAAACACAACGATGATG	hsp90ab1
Hepcidin (*hamp*)	Ch_aur008.1	101–200	AGGAGGCAGGGAGCAATGACACTCCAGTTGCGGCACATCAAGAAATGTCAATGGAATCGTGGATGATGCCGAGTCGCGTCAGGGAGAAGCGTCAGAGCCA	hamp
Mitochondrial uncoupling protein 2-like (*ucp2-like*)	Ch_aur038.1	539–638	TCACTAGAAATGCGCTTGTCAACTGCACAGAACTGGTTACATACGACCTGATCAAGGAGGCCATCCTCAAACACAACCTGTTGTCAGACAACCTGCCCTG	LOC115579854
Nuclear factor erythroid 2-related factor 2 (*nrf2*)	Ch_aur065.1	1360–1459	CAGAGGGCTAAGGCCCTCAAAATCCCTTTCACTGTAGACATGATTATCAATCTGCCTGTCGACGATTTCAATGAGCTGATGTCAAAGCACCGACTGAATG	nfe2l2
Peroxiredoxin-1 (*prdx1*)	Ch_aur042.1	162–261	CGAGATCATAGCTTTCAGTGACGCTGCTGACGATTTCAGGAAGATCGGCTGTGAGGTCATCGCCGCCTCTGTTGACTCACACTTCTCCCATTTCGCATGG	LOC115573364
Peroxiredoxin-1-like (*prdx-like*)	Ch_aur013.1	374–473	CATACAGGGGGCTGTTTGTGATTGACGACAAGGGCATCTTGAGGCAGATCACCATCAATGACTTGCCTGTGGGTCGCTCTGTGGATGAGACTCTGCGCCT	LOC115587998
Peroxiredoxin-5 mitochondrial (*prdx5*)	Ch_aur057.1	152–251	TGTCTATGGATCAGCTCTTCAAGGGGAAGAAGGGAGTCCTCTTTGCTGTACCTGGAGCCTTCACACCTGGATGTTCCAAGACTCACCTCCCAGGTTTTGT	prdx5
Serotransferrin-like (*tf-like*)	Ch_aur005.1	582–681	CGAGCCTTATTATGACTACGGTGGAGCCTTCCAATGTCTGGCAGACGACGCTGGTGATGTGGCCTTTGTGAAGCATCTCACTGTACCTGAGTCTGAAAAG	LOC115572354
Superoxide dismutase [Cu-Zn] (*sod1*)	Ch_aur059.1	100–199	GGAGAAATCTCGGGACTTACTCCTGGTGAGCATGGTTTCCATGTCCATGCATTTGGAGACAATACAAATGGGTGCATCAGTGCAGGCCCTCACTTCAATC	sod1
Thioredoxin-dependent peroxide reductase mitochondrial (*prdx3*)	Ch_aur006.1	290–389	CCTTTGTGTGTCCAACAGAGATCATCTCATTCAGCGACAAGGCCAGTGAGTTCCACGACGTTAACTGTGAGGTGGTGGGTGTGTCGGTGGACTCTCACTT	prdx3
Genes that did not pass calibration				
Alkaline phosphatase tissue-nonspecific isozyme isoform X1 (*alpl*)	Ch_aur035.1	661–760	GGCTGCAAGGATATCGCCAGACAACTCTTTGAAAATATTCCCAACATTGATGTGATTATGGGTGGAGGAAGGAAGTATATGTTCCCTAAAAACAAGTCGG	alpl
Cell surface glycoprotein MUC18-like isoform X (*muc18-like*)	Ch_aur003.1	412–511	ACTTACTTTGTTCCTGGAGGAACCAGGATGACTGAGACCAACCGTATTAACATCACTGTATACTACCCCTCCACCGCTGTAAGTGTTTGGGTGGAGTCAC	LOC115570482
Complement component C6 (*c6*)	Ch_aur018.1	2413–2512	CTCTGTATCCTGAACGTAGACCTCGGCGTCACCGTGCCGATGTCCCTCTGCTCCTTCCACGTCGGGCTTTGCCACAATGATCCGCTCTTCTATGTCAGCG	c6
Complement component C8 alpha chain (*c8a*)	Ch_aur047.1	568–667	TGGAGGAAATTCAGCTATGACTCATTCTGTGAGAACCTGCACTACAATGAAGATGAGAAGAACTACAGGAAACCTTACAACTACCACACCTACCGTTTTG	c8a
Glucocorticoid receptor-like (*gcr-like*)	Ch_aur049.1	669–768	GGACGTTGGCTCAGAGAGGGACATGAAGTCTGCTGTGGTTGAAAGCATTAACGGCAGTGGGGCAGTCTTTGTTGCTCTTAATGGCAGTAATATGACAAGT	LOC115568693
Glutathione peroxidase 2 (*gpx2*)	Ch_aur064.1	103–202	TGTGGCCTCGCTCTGAGGCACCACCACCCGGGACTACAGCGAGCTCAACCAGCTGCAGAGCAAGTACCCGCATCGGCTGGTGGTCCTGGGTTTTCCCTGT	gpx2
Insulin-like growth factor I isoform X1 (*igf1*)	Ch_aur033.1	188–287	GAGAGAGAGGCTTTTATTTCAGTAAACCTGGCTATGGCCCCAATGCACGGCGGTCACGTGGCATTGTGGACGAGTGCTGCTTCCAAAGCTGTGAGCTGCG	igf1
Insulin-like growth factor II (*igf2*)	Ch_aur050.1	161–260	CGCTGTGTGGGGGAGAGCTGGTGGATGCGCTGCAGTTTGTCTGCGAAGACAGAGGCTTCTATTTCAGTAGGCCAACCAGCAGGGGAAACAACCGGCGCCC	igf2
Interferon-induced GTP-binding protein Mx-like B (*mx-like*)	Ch_aur051.1	982–1081	CCATCTGATGCAGCTGAGAGAGTCGTCTTCCTCATTGATAAAGTGACAGCTTTCACTCAGGATGCCATCAGTCTGACCACAGGAGAAGAACTCAATTGTG	LOC115583120
Interleukin-1 beta-like (*il1b-like*)	Ch_aur029.1	412–511	CCTACACCCAGTGCTGAGGCCGTAACTGTGACTCTGTGCATCAAGGACACAAATCTTTACCTGTCTTGTCACAAGGAAGGTGACGAGCCAACCTTGCATC	LOC115581181
Interleukin-6 isoform X1 (*il6*)	Ch_aur045.1	127–226	GTGATGCTGGCCGCTCTGCTTCAGTGTGCTCCCGGGGCTCCGATTGATGGCGCGCTCACTGACAATCCAGCAGGTGACACCTCAGGTGAAGAGTGGGAGA	LOC115579128
Interleukin-10-like (*il10-like*)	Ch_aur062.1	528–627	AGGTCTATACAAGGCCATGGGAGAGCTGGATCTGCTGTTCAACTACATTGAGACATATCTGGCTTCCAAACGGCACGGAACACATGTGGCCTCCGCTTGA	LOC115582730
Interleukin-12 subunit beta-like (*il12b-like*)	Ch_aur032.1	388–487	GCACCTAACTATTCAGGCTCCTTCAAATGCACCTGGGCTAAAGCAGAGCACAGATCCAACGCCGCCGTGCTCCTGGTGAAGGCCGAACGTCATTTGGAGA	LOC115593944
Interleukin-17D (*il17d*)	Ch_aur010.1	415–514	CGCAGCACTCCGGTCTACGCTCCGTCTGTCATCCTGAGGAGAACCGGCTCCTGCCTCGGCGGCCGACACTCATACACAGAGATCTACGTCTCCATCGCGG	il17d
Interleukin-34 isoform X1 A (*il34*)	Ch_aur009.1	142–241	CGGTACATGAGGCACAACTTCCCCATCAAGTACACCATCAGGGTTCATCACAACGAAGTCTTTAAACTGTCAAACATCAGCAGACTGAGGTTACAGGTGG	il34
Macrophage colony-stimulating factor 1 receptor (*csf1r*)	Ch_aur060.1	2343–2442	CAAAAATTGTATTCACAGAGACATCGCTGCAAGGAATGTCCTGTTGACTGATCACAGAGTGGCCAAGATTTGTGACTTTGGTCTGGCACGTGACATCATG	csf1r
Mucin-2-like isoform X1 (*muc2-like*)	Ch_aur020.1	1740–1839	CTGTTCCCTCAGTGTGGAAAATGAGAATTACGCCAAACACTGGTGTGCCTTGCTGCTAAGTCCAGACAGCTCCTTTGCACAGTGCCGTTCAGCGGTGGAT	LOC115586438
Nuclear factor NF-kappa-B p100 subunit isoform X1 (*nfkb2*)	Ch_aur030.1	717–816	GGAGGCGTTCGGAGACTTTTCACCAACCGACGTTCACAAACAGTACGCCATTGTGTTCAAAACGCCGCCCTATCACAGCGCAGAGATCGAGCGGCCCGTC	nfkb2
Prostaglandin G/H synthase 2 (*ptgs2*)	Ch_aur071.1	1187–1286	TCGTCTTCAACACGTCTGTAGTGACTGAGCACGGCATCAGCAACCTTGTGGAGTCGTTTTCCAAGCAGATCGCTGGACGGGTTGCCGGTGGCCGAAATGT	ptgs2
Stromal cell-derived factor 1 (*cxcl12*)	Ch_aur016.1	197–296	AGAACAACAGGGAAGTTTGCATCAACCCGGAGACCAAGTGGCTGCAGCAGTACTTAAAGAACGCCATTAACAAGGTGAAGAAAAACCGAAGACGCAATAA	cxcl12
Suppressor of cytokine signalling 3 (*socs3*)	Ch_aur039.1	291–390	GCGCATCCAGTGTGACTCAAGCTCTTTTTTCCTGCAGACGGACCCTAAAAACGTTCAGTCTGTTCCTCACTTTGACTGCGTCCTCAAGCTGGTGCATTAC	socs3
T-cell surface glycoprotein CD8 alpha chain-like isoform X3 (*cd8a-like*)	Ch_aur017.1	151–250	TGGTTTCGAGTGCTGGACAAATCTGGCATGGAATTCATTGGGTCTTTCAGCAATACTGGCGTGAAAAAACCAAATACAAAGCCTCCAACTTCCATCTTCA	LOC115583354
Toll-like receptor 2 isoform X1 (*tlr2*)	Ch_aur048.1	602–701	CGAGGTATGAGTCCGGTACTCTGGCATACGTTTGGCCGTTGGGTCGTGTCACTTTGAGCCTCCACAGTCCATTTTTAACAAATGAGGCCTTAGCCTCAGC	LOC115590525
Toll-like receptor 3 (*tlr3*)	Ch_aur014.1	799–898	AGCCAAGCTGATGGCAGCTTTCAGCCGTACAGCGCGGTGCTGCAGACCACTGAATCACTCAAAGTACTTCAGCTGCAATTCATGAAGGTGTTGATAGAAA	LOC115590587
Toll-like receptor 5 (*tlr5*)	Ch_aur053.1	1203–1302	CTTCCCTGCGTCTCTACCCAGATTAGATTATCTCCTGTTGAACGACAACAAGTTGACCTCCTCGTCAGTATACAGTCTCACACGGTTTGCCGATAATGCC	LOC115574263
Transforming growth factor beta-1 proprotein-like isoform X1 (*tgfb1-like*)	Ch_aur024.1	300–399	CAGTGCCATCAATTTTGAGGTCTCCGGGATCTCGAATAGTAGGGGAGACACACAAGGGTTTCAACAGGTGTCGCAGCAATACCCGTACATCCTGACCATG	LOC115575711

## Data Availability

The data generated and analysed in this study are available upon request from the corresponding author. Access to these data is contingent upon obtaining appropriate consent from the guardians (kaitiaki) of snapper, the Māori indigenous people of New Zealand, to honour their role as stewards of this taonga (treasured species). Potential requestors must provide a detailed explanation of their intended research purpose, including the potential outcomes and benefits of their work. Additionally, requestors are expected to outline any benefit-sharing arrangements before data access can be requested. This ensures that the use and reuse of the data respects the cultural values and guardianship of Māori over natural resources.

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
