# Peer review of "Assessment of a Novel Stress and Immune Gene Panel on the Development of Australasian Snapper (Chrysophrys auratus) Larvae"

_genes, 2024, doi:10.3390/genes15121520_

Round 1
Reviewer 1 Report
Comments and Suggestions for Authors
MS contains valuable studies on the larval development of Australasian snapper and analyses of the expression of genes related to stress and immune responses. This study addresses a key stage in aquaculture, as larval development and their ability to respond to stress factors and immune threats are crucial to their survival and health.
This study is described as the first to analyse the expression of a panel of stress and immune-related genes in snapper larvae. The introduction of this methodology provides a basis for further studies that may expand our knowledge of the effects of different environmental factors on early fish development.
This study is well-founded and provides valuable data on the genetics of Australasian snapper larval development. It fills a gap in knowledge of early immune and stress responses in fish and offers potential guidance for future studies on larval development in aquaculture. This work could have important implications not only for the development of Australasian snapper aquaculture but also for the general understanding of the adaptive mechanisms of fish larvae in response to stress, which may have application in the culture of other fish species.
However, I believe that the MS contains many errors and is unfinished. My detailed comments are below.
The weakest point of this MS is the Discussion section. The authors describe their research, but do not refer to the results of other authors. There is no comparison with other fish larvae or older individuals. There is no explanation of the conclusions the authors reached by discovering so many genes. The authors did not provide an answer to the purpose of the work.
There is no comparison with other studies that described the expression of genes related to immunity and stress response in fish, including larvae of different species. For example, studies on salmon and trout larvae have shown that genes related to the response to oxidative stress and the immune response are crucial in the first days of development, similarly to the described study of Australasian snapper.
Specific comments:
Lines 13-14: This sentence is not understandable "Larvae were collected between 1145 h and 1610 h each day" Did you mean 11:45 AM to 16:10 PM? Please correct it.
Line 18: Replace ''Here we'' with "In this study"
Line 24: Please add some more keywords that are different from the MS title
Line 58: Remove "We" and start the sentence with: Heve
Line 72: Replace ''Here we'' with "In this study"
Line 120: The description of the figure should be more precise. Please add the English and Latin name of the species. Please move "a" and "b" to the end of the description of each fig. For example: Figure 1. The funnel and mesh arrangement used to capture the eggs/larvae (a).
Line 123: "concentration of 5 ml per 1000L" Isn't that a mistake? Please check.
Line 123: Were any studies done on fish larvae using this anesthetic to determine the optimal doses? Please add references.
Lines 135-136: Please change this sentence. I suggest: "Details of the literature searches and design process were applied according to Bentley-Hewitt et al. [13]."
Lines 176-188: This fragment is not results. Please move it to the Materials and Methods section.
Line 196: The description of Figure 2 is incorrect. Please be clear about what Fig. 2 shows. Please include the English and Latin species name in the description.
The "Y" axis is not labeled. At least the first Fig. on the left should have a labeled axis.
Line 205: PCA analysis without a table is incomplete. Please add a table that presents correlations of at least 3 - 4 loadings. Dim1 end Dim2 explains only about 51%, therefore I propose adding values for Dim3 and Dim4 in the table. The description of Fig. should include the English and Latin species name.
Line 228: Please correct the name of Fig. 4. The description of the figure should be precise and should include the Latin and English name of the species on which the research was performed.
Line 229. References should be appropriately formatted. References should include chapter and page numbers. Latin names of species should be italicized.
Author Response
|
Response to Reviewer 1 Comments
|
||||||||||||||||||||||||||||||||||||||||||||||||||||||||||||||||||||||||||||||||||||||||||||||||||||||||||||||||||||
|
1. Summary |
|
|
||||||||||||||||||||||||||||||||||||||||||||||||||||||||||||||||||||||||||||||||||||||||||||||||||||||||||||||||||
|
Thank you very much for taking the time to review this manuscript. Please find the detailed responses below and the corresponding revisions/corrections highlighted/in track changes in the re-submitted files.
|
||||||||||||||||||||||||||||||||||||||||||||||||||||||||||||||||||||||||||||||||||||||||||||||||||||||||||||||||||||
|
Point-by-point response to Comments and Suggestions for Authors |
||||||||||||||||||||||||||||||||||||||||||||||||||||||||||||||||||||||||||||||||||||||||||||||||||||||||||||||||||||
|
Comments 1: MS contains valuable studies on the larval development of Australasian snapper and analyses of the expression of genes related to stress and immune responses. This study addresses a key stage in aquaculture, as larval development and their ability to respond to stress factors and immune threats are crucial to their survival and health.
This study is described as the first to analyse the expression of a panel of stress and immune-related genes in snapper larvae. The introduction of this methodology provides a basis for further studies that may expand our knowledge of the effects of different environmental factors on early fish development.
This study is well-founded and provides valuable data on the genetics of Australasian snapper larval development. It fills a gap in knowledge of early immune and stress responses in fish and offers potential guidance for future studies on larval development in aquaculture. This work could have important implications not only for the development of Australasian snapper aquaculture but also for the general understanding of the adaptive mechanisms of fish larvae in response to stress, which may have application in the culture of other fish species. Response 1: We appreciate your positive response regarding the importance of the study.
Comments 2: However, I believe that the MS contains many errors and is unfinished. My detailed comments are below. The weakest point of this MS is the Discussion section. The authors describe their research, but do not refer to the results of other authors. There is no comparison with other fish larvae or older individuals. There is no explanation of the conclusions the authors reached by discovering so many genes. The authors did not provide an answer to the purpose of the work. Response 2: Thank you for your comments. We have added more information into the discussion to compare larvae gene expression to older fish and other authors results. A new sections to the discussion which reads “Juvenile snapper fin, head kidney and liver were analyzed with the panel of genes pre-sented in this study and 6 more genes passed calibration (muc18-like, c8a, igf1, ptgs2, soc3 and tgfb1-like), indicating that expression of those genes were very low or not present in snapper larvae in the first 30 days of development [13]. For example, muc18-like, MUC18 is a predominant mucin in skin, gills and stomach in gilthead seabream and offers a first line of defence [20] and therefore may not be being produced at high concentrations whilst snapper are in early development stages. Another first line of defense is the complement system. C8A is a terminal complement protein, which forms the membrane attack com-plex sand can lyse certain pathogens and cells [21]. Since c8a and muc18-like are both not present or not highly expressed in larvae demonstrates that these immune defenses may not be well established in the first 30 days of larvae development. Although another com-ponent of the complement system, C3-like, was found in larvae in this study, indicating that the system is at least partly present. A study in Eurasian perch larvae demonstrated difference in gene expression of C3 when comparing domesticated (higher expression) to wild Eurasian perch in early developmental stages (18-20 day post hatch) and domesticated fish coped better with thermal stress [22]. This indicates the potential importance of this gene in the tolerance of larvae to thermal stress.” Lines 262-279. In addition, we have included more information regarding the conclusions and purpose of the work in lines 310-318 “Future work should demonstrate which gene markers are useful to predict stress or disease responses in the larvae and what stressors cause the most significant effects on gene changes. This approach could potentially be used as a fish health husbandry management tool. In conclusion, this study is the first to explore immune and stress related genes in snapper larvae. It indicates what components of the immune system and stress response related genes are present in very early development of snapper larvae. Understanding these processes is critical to understanding fish health and can aid the development of good aquaculture practices for the species.”
Comments 3: There is no comparison with other studies that described the expression of genes related to immunity and stress response in fish, including larvae of different species. For example, studies on salmon and trout larvae have shown that genes related to the response to oxidative stress and the immune response are crucial in the first days of development, similarly to the described study of Australasian snapper. Response 3: We have included another comparison with a gene expression study in larvae from an alternative species. Lines 273-279 “A study in Eurasian perch larvae demonstrated difference in gene expression of C3 when comparing domesticated (higher expression) to wild Eurasian perch in early develop-mental stages (18-20 day post hatch) and domesticated fish coped better with thermal stress [22]. This indicates the potential importance of this gene in the tolerance of larvae to thermal stress.” This is in addition to studies in gilthead seabream larvae.
Specific comments: Comments 4: Lines 13-14: This sentence is not understandable "Larvae were collected between 1145 h and 1610 h each day" Did you mean 11:45 AM to 16:10 PM? Please correct it. Response 4: Thank you for pointing this out. The changes have been made throughout the manuscript.
Comments 5: Line 18: Replace ''Here we'' with "In this study" Response 5: Thank you for pointing this out. The changed have been made on line 12
Comments 6: Line 24: Please add some more keywords that are different from the MS title Response 6: We have included additional keywords “stress resistance and early snapper development” on lines 24-25.
Comments 7: Line 58: Remove "We" and start the sentence with: Heve Response 7: The word “we” was removed and replace with “Research at Plant & Food has included the development of….” On line 59
Comments 8: Line 72: Replace ''Here we'' with "In this study" Response 8: Thank you for pointing this out. The changes have been made on line 74
Comments 9: Line 120: The description of the figure should be more precise. Please add the English and Latin name of the species. Please move "a" and "b" to the end of the description of each fig. For example: Figure 1. The funnel and mesh arrangement used to capture the eggs/larvae (a). Response 9: Thank you for pointing this out the new figure legend reads “The funnel and mesh arrangement used to capture the “dry” eggs/larvae of Chrysophrys auratus, tāmure, (family Sparidae, referred to as snapper (a). The pre-weighed patch of mesh with collected snapper eggs (b).” on lines 123-125.
Comments 10: Line 123: "concentration of 5 ml per 1000L" Isn't that a mistake? Please check. Response 10: Thank you for pointing this out this is a mistake and has been corrected to “40 mg/L” on line 127
Comments 11: Line 123: Were any studies done on fish larvae using this anesthetic to determine the optimal doses? Please add references. Response 11: Aqui-S dose (40 mg/L) is a euthanasia overdose for fish in general (reference https://doi.org/10.1111/jfb.12658 for early-juvenile barramundi). For larvae, we normally euthanise in at least 2 steps aiming for ~10 mg/L first, and then full dose once fish are anesthetised. We have added this reference into the manuscript.
Comments 12: Lines 135-136: Please change this sentence. I suggest: "Details of the literature searches and design process were applied according to Bentley-Hewitt et al. [13]." Response 12: Thank you for pointing this out. This change has been made on Line 141. Also note that the reference has been updated now the article is in press.
Comments 13: Lines 176-188: This fragment is not results. Please move it to the Materials and Methods section. Response 13: Thank you for pointing this out. This section has been moved to the methods section and re-labeled section 2.5 on lines 172-183. Section numbers have been updated to reflect this change.
Comments 14: Line 196: The description of Figure 2 is incorrect. Please be clear about what Fig. 2 shows. Please include the English and Latin species name in the description. The "Y" axis is not labeled. At least the first Fig. on the left should have a labeled axis. Response 15: The figure legend has been updated to include the English, Māori and Latin name of the species on lines 216-217. Figure 2 now includes the y axis.
Comments 16: Line 205: PCA analysis without a table is incomplete. Please add a table that presents correlations of at least 3 - 4 loadings. Dim1 end Dim2 explains only about 51%, therefore I propose adding values for Dim3 and Dim4 in the table. The description of Fig. should include the English and Latin species name. Response 16: The figure legend has been updated to include the English, Māori and Latin name of the species on lines 229-230. We have included a supplementary table shown below. This is referred to in lines 224-246.
Supplementary Table 1. Showing how variables correlate with the first 5 principal components (PCs). This accounts for 79% variance in total.
Summary PC3 seems associated with some oscillations in the balance of GSR and NRF2 vs PRDX1-like over a 8-10 cycle; GRS and NRF2 are high or PRDX1 is low initially (Day 0 and 4), on Day 14 and on Days 22 and 24. On Days 6, 18 and 28 PRDX1 is higher and/or GRS and NRF2 are lower. PC4 seems driven by a change that occurs at Day 8, compared to Days 0-4 or Day 14 onwards; ACTB, HSP70 and PRDX1-like are low, PRDX3 and PRDx5 are higher.
Comments 17: Line 228: Please correct the name of Fig. 4. The description of the figure should be precise and should include the Latin and English name of the species on which the research was performed. Response 17: The figure legend has been updated to include the English, Māori and Latin name of the species on lines 252-253.
Comments 18: Line 229. References should be appropriately formatted. References should include chapter and page numbers. Latin names of species should be italicized. the full reviewer comment here.] Response 18: Thank you for pointing this out. Edits have been made to the reference section.
|
||||||||||||||||||||||||||||||||||||||||||||||||||||||||||||||||||||||||||||||||||||||||||||||||||||||||||||||||||||
Reviewer 2 Report
Comments and Suggestions for Authors
Paper "Assessment of a novel stress and immune gene panel on the development of Australasian snapper (Chrysophrys auratus) larvae" is a significant contribution to understanding the biology of the development of this promising aquaculture species. Generally, I believe that research on larvae represents an extremely important contribution to aquaculture studies because larval stages are a short but crucial phase in the life of fish. During the larval period, many changes occur that subsequently affect the lives of adult individuals, which in turn relates to the profitability of the entire production.
However, I find that the presented work is lacking several pieces of information regarding the larval period, specifically the weight and length of larvae during sampling points and the mortality rate of the larvae. These zootechnical parameters are important indicators of the "correctness" of the larval rearing, the overall health of the fish, and their welfare. In the search for stress and immune markers, this becomes even more important.
I also miss a few sentences providing a real summary of the work. The last paragraph of the discussion mentions limitations and potential studies that need to be undertaken in the future, but it does not summarize what the Authors want us as readers to take away as the 'take-home message.'
On the other hand, I really like the in-depth exploration of the timing of sample collection and the attempt to find the best moment.
Below are some detailed comments.
Line 52-53: Looking at the other citations in the text, it seems this one should also be in the form of a number.
Line 64: I think only the number '13' should appear here as the citation.
Line 75: I think it would look better here as: Egg collection, larvae growth, and rearing. This would represent a sequence of events in line with the 'timeline’.
Line 94: Could Authors provide what is the natural photoperiod for this fish at that time of experiment?
Line 117: Could the Authors provide the sizes of the larvae at the time of sampling? This would allow for the observation of the dynamics of larvae growth.
Line 128: I suggest just write – This was done in triplicates.
Line 136: Same as in line 64
Line 138: Could the Authors provide how many larvae were included in the 25 mg for each sampling? I understand that at the beginning there were definitely more larvae than at the end of the rearing process. For example, if at the start it was a homogenate of, say, 10 larvae, could it be that at the end, those 25 mg represented a single larva?
Line 178: Same as in lines 64 and 136
Line 191: Generally, this is not a significant error; however, it has become common practice that in the case of fish, gene symbols are written either in italics with uppercase letters (e.g., Sod1) or in italics with lowercase letters (e.g., sod1). Uppercase letters are usually reserved for human genes, while in the case of fish, they are often used for proteins.
Figure 2: Did Authors observed any significant differnces in case of presentetd genes between sampling days? Would be good to provide this data on the picture that reader have idea that for example in the case of hsp70 there were differnces between Day 2 and 6?
Line 234-235: I do not agree with the Authors that a homogenate of the entire larva could be a limiting factor. I believe that research on larvae gives us the incredible opportunity to trace what is happening at the level of the whole organism and all tissues (which is very beneficial in the case of the presented studies), something we cannot achieve with such minimal effort in older fish.
Author Response
|
Response to Reviewer 2 Comments
|
||
|
1. Summary |
|
|
|
Thank you very much for taking the time to review this manuscript. Please find the detailed responses below and the corresponding revisions/corrections highlighted/in track changes in the re-submitted files.
|
||
|
Point-by-point response to Comments and Suggestions for Authors |
||
|
Comments 1: Paper "Assessment of a novel stress and immune gene panel on the development of Australasian snapper (Chrysophrys auratus) larvae" is a significant contribution to understanding the biology of the development of this promising aquaculture species. Generally, I believe that research on larvae represents an extremely important contribution to aquaculture studies because larval stages are a short but crucial phase in the life of fish. During the larval period, many changes occur that subsequently affect the lives of adult individuals, which in turn relates to the profitability of the entire production. Response 1: We appreciate your positive response regarding the importance of the study.
Comments 2: However, I find that the presented work is lacking several pieces of information regarding the larval period, specifically the weight and length of larvae during sampling points and the mortality rate of the larvae. These zootechnical parameters are important indicators of the "correctness" of the larval rearing, the overall health of the fish, and their welfare. In the search for stress and immune markers, this becomes even more important. Response 2: Thank you for your comment. This study was initially designed as a pre-liminary study to test whether a panel of genes designed for juvenile snapper would work for snapper larvae. We do not have data for larvae length; however, 25 mg of larvae was consistently used for the gene expression analysis. The larvae used for the study were all living at the time of sampling prior to euthanasia. We have included a supplementary figure with images of day 2 and day 30 larvae with measurements (shown below). This is referred to in the text on line 135-136.
Supplementary figure 1. Images of snapper at day 2 (post hatch) and day 30. (Image shown in attached document)
Comments 3: I also miss a few sentences providing a real summary of the work. The last paragraph of the discussion mentions limitations and potential studies that need to be undertaken in the future, but it does not summarize what the Authors want us as readers to take away as the 'take-home message.' Response 3: Thank you for pointing this out. We have included extra information regarding the conclusions and purpose of the work in lines 310-318 “Future work should demonstrate which gene markers are useful to predict stress or disease responses in the larvae and what stressors cause the most significant effects on gene changes. This approach could potentially be used as a fish health husbandry management tool. In conclusion, this study is the first to explore immune and stress related genes in snapper larvae. It indicates what components of the immune system and stress response related genes are present in very early development of snapper larvae. Understanding these processes is critical to understanding fish health and can aid the development of good aquaculture practices for the species.”
Comments 4: On the other hand, I really like the in-depth exploration of the timing of sample collection and the attempt to find the best moment. Response 4: Thank you for this positive comment on our study.
Below are some detailed comments. Comments 5: Line 52-53: Looking at the other citations in the text, it seems this one should also be in the form of a number. Response5: This reference has been removed from the text as there is a reference for this species at the end of the sentence. Line 54.
Comments 6: Line 64: I think only the number '13' should appear here as the citation. Response 6: Thank you for pointing this out. This reference has now been published and full details are included in the reference list.
Comments 7: Line 75: I think it would look better here as: Egg collection, larvae growth, and rearing. This would represent a sequence of events in line with the 'timeline’. Response 7: Thank you for pointing this out the text has been updated. Line 78.
Comments 8: Line 94: Could Authors provide what is the natural photoperiod for this fish at that time of experiment? Response 8: Thank you for pointing this out. This has now been included on line 98 “(15:9 Light: Dark).”
Comments 9: Line 117: Could the Authors provide the sizes of the larvae at the time of sampling? This would allow for the observation of the dynamics of larvae growth. Response 9: Thank you for your comment. This study was initially designed as a pre-liminary study to test whether a panel of genes designed for juvenile snapper would work for snapper larvae. We do not have data for larvae size; however, 25 mg of larvae was consistently used for the gene expression analysis. However, we have included a supplementary figure which shows images and lengths of day 2 and day 30 snapper larvae. This is referred to in lines 135-136.
Comments 10: Line 128: I suggest just write – This was done in triplicates. Response 10: Thank you for pointing this out. The text has been updated. Lines 132-133.
Comments 11: Line 136: Same as in line 64 Response 11: Thank you for pointing this out. This reference has now been published and full details are included in the reference list.
Comments 12: Line 138: Could the Authors provide how many larvae were included in the 25 mg for each sampling? I understand that at the beginning there were definitely more larvae than at the end of the rearing process. For example, if at the start it was a homogenate of, say, 10 larvae, could it be that at the end, those 25 mg represented a single larva? Response 12: Thank you for your comment. This study was initially designed as a pre-liminary study to test whether a panel of genes designed for juvenile snapper would work for snapper larvae. We do not have data for the exact number of larvae used for each sampling day of the standardized weight of 25 mg of larvae. However, there was a minimum of 1 whole homogenized larva in each sample. In our lab book records it is noted that approximately 10-20 eggs/larvae were included between day 0- day 10 and on day 16 there were 10 larvae.
Comments 13: Line 178: Same as in lines 64 and 136 Response 13: Thank you for pointing this out. This reference has now been published and full details are included in the reference list.
Comments 14: Line 191: Generally, this is not a significant error; however, it has become common practice that in the case of fish, gene symbols are written either in italics with uppercase letters (e.g., Sod1) or in italics with lowercase letters (e.g., sod1). Uppercase letters are usually reserved for human genes, while in the case of fish, they are often used for proteins. Response 14: Thank you for pointing this out. The text has been updated throughout the manuscript.
Comments 15: Figure 2: Did Authors observed any significant differnces in case of presentetd genes between sampling days? Would be good to provide this data on the picture that reader have idea that for example in the case of hsp70 there were differnces between Day 2 and 6? Response 15: We collected samples every 2 days throughout the study and all gene expression data is included in figure 2.
Comments 16: Line 234-235: I do not agree with the Authors that a homogenate of the entire larva could be a limiting factor. I believe that research on larvae gives us the incredible opportunity to trace what is happening at the level of the whole organism and all tissues (which is very beneficial in the case of the presented studies), something we cannot achieve with such minimal effort in older fish. Response 16: Thank you for your comment. We were attempting to highlight that there could be differences in gene expression in different sections of the larvae once organs start to develop. We previously found in juvenile snapper that there was a lot of variances in gene expression in the fin, head kidney and liver.
|
||

Reviewer 3 Report
Comments and Suggestions for Authors
This is a baseline gene expression study of larval snapper species aimed at understanding the development of stress and immune responses to potential environmental and nutritional changes in the larval culture stages of the fish.
The paper is well represented and included -
a. A background review summary of current gene expression studies for the fish species.
b. The experimental design is well considered and a controlled environment with sufficient replicates of the tissue samples collected.
c. The results indicated a pattern of responses for most gene targets related to stress and immune function being active across the 30 day study period.
d. As this is a baseline 'normal' study, no inciting stress or pathogen challenge was designed into the study.
e. The conclusion of the study is reasonable, indicating that the larvae's stress and immune gene targets are responsive.
Future work will hopefully demonstrate which gene markers are useful to predict stress or disease responses in the larvae and what stressors cause the highest responses. This approach may potentially be used as a fish health husbandry management tool.

Author Response
|
Response to Reviewer 3 Comments
|
||
|
1. Summary |
|
|
|
Thank you very much for taking the time to review this manuscript. Please find the detailed responses below and the corresponding revisions/corrections highlighted/in track changes in the re-submitted files.
|
||
|
Point-by-point response to Comments and Suggestions for Authors |
||
|
Comments 1: This is a baseline gene expression study of larval snapper species aimed at understanding the development of stress and immune responses to potential environmental and nutritional changes in the larval culture stages of the fish.
The paper is well represented and included -
a. A background review summary of current gene expression studies for the fish species.
b. The experimental design is well considered and a controlled environment with sufficient replicates of the tissue samples collected.
c. The results indicated a pattern of responses for most gene targets related to stress and immune function being active across the 30 day study period.
d. As this is a baseline 'normal' study, no inciting stress or pathogen challenge was designed into the study.
e. The conclusion of the study is reasonable, indicating that the larvae's stress and immune gene targets are responsive.
Future work will hopefully demonstrate which gene markers are useful to predict stress or disease responses in the larvae and what stressors cause the highest responses. This approach may potentially be used as a fish health husbandry management tool.
|
||
|
Response 1: Thank you for your positive comment on our study. We appreciate you highlighting future work that this study could be used for and we have included this in our discussion.
|
||
Round 2
Reviewer 1 Report
Comments and Suggestions for Authors
MS has been significantly improved. I thank the authors for taking my comments into account. The only comment I have is about the figures (especially PCA), which could be of better quality. I suggest you enlarge the figures.
Author Response
|
1. Summary |
|
|
Thank you very much for taking the time to review this manuscript again.
|
|
|
Comments 1: MS has been significantly improved. I thank the authors for taking my comments into account. The only comment I have is about the figures (especially PCA), which could be of better quality. I suggest you enlarge the figures. Response 1: Thank you for your positive comments on improving the manuscript. We have uploaded additional files for figure 2-4 with improved quality.
Additionally, we have made some minor edits to the manuscript in the attached document shown in track changes.
|
|
